# The Role of Inflammation in Anal Cancer

**DOI:** 10.3390/diseases10020027

**Published:** 2022-05-06

**Authors:** Amir Selimagic, Ada Dozic, Azra Husic-Selimovic, Nijaz Tucakovic, Amir Cehajic, Anela Subo, Azra Spahic, Nedim Vanis

**Affiliations:** 1Department of Gastroenterohepatology, General Hospital “Prim. dr. Abdulah Nakas”, 71 000 Sarajevo, Bosnia and Herzegovina; husic_azra@yahoo.com (A.H.-S.); tucakovic.nijaz@gmail.com (N.T.); amircehajic@hotmail.com (A.C.); 2Department of Internal Medicine, General Hospital “Prim. dr. Abdulah Nakas”, 71 000 Sarajevo, Bosnia and Herzegovina; ada.dozic@gmail.com (A.D.); anelasubo@gmail.com (A.S.); azraspahicc@gmail.com (A.S.); nedim_08@yahoo.com (N.V.)

**Keywords:** anal cancer, inflammation, HPV, HIV, inflammatory bowel disease

## Abstract

The aim of this article was to present a summary of the current resources available in the literature regarding the role of inflammation in anal cancer development. Anal cancer is relatively uncommon, accounting for about 2.7% of all reported gastrointestinal cancers in the United States. However, the importance of understanding the pathogenesis and risk factors for anal cancer has been recognized over the last several decades due to a noticed increase in incidence worldwide. Infections, autoimmune diseases, and inflammatory diseases of unknown etiology cause chronic inflammation that promotes tumorigenesis. The association between chronic inflammation and cancer development is widely accepted. It is based on different pathophysiological mechanisms that lead to cellular transformation and changes in immunological response, allowing tumor cells to avoid apoptosis and immune surveillance. However, there are still many molecular and cellular mechanisms that remain largely unexplored. Further studies on this topic could be of tremendous significance in elucidating anal cancer pathogenesis and developing immunotherapeutic approaches for its treatment.

## 1. Introduction

Anal cancer is relatively uncommon, accounting for about 2.7% of all reported gastrointestinal cancers in the United States (US) [1]. In 2022, it is expected that there will be 9440 new cases and 1670 deaths due to anal cancer [1].

The anal canal is a short (3–4 cm) segment located distal to the rectum, commencing at the anorectal ring and extending to the anal verge. There are different types of epithelium located in this region, including, from proximal to distal: (1) colorectal-like columnar epithelium; (2) stratified columnar epithelium, which is similar to transitional epithelium of the genitourinary tract; (3) non-keratinized squamous cell epithelium [2]. A hair-bearing squamous epithelium commences at the anal verge, and the region located within 5 cm of the anal verge, is defined as perianal skin [2]. The tumors located in anal region are classified as anal canal and anal margin cancers [2]. The World Health Organization classifies anal canal cancers as epithelial, mesenchymal and secondary tumors (Appendix A) [3]. Anal squamous cell cancer (SCC) occurs most frequently, accounting for approximately 80% to 85% of all anal canal cancers [4].

The importance of understanding the pathogenesis and risk factors for anal cancer has been recognized over the last several decades due to an increase in incidence that has been noticed worldwide.

Human papillomavirus (HPV) infection, female sex, total lifetime number of sexual partners, a history of sexually transmitted viral infections, cigarette smoking, receptive anal intercourse, and HIV infection are all risk factors for anal cancer [5]. Patients taking immunosuppressive therapy, e.g., after organ transplantation, have an increased risk of developing anal cancer as well. Risk factors related to female sex include history of high grade vulvar intraepithelial neoplasia, vulvar or cervical cancer [5].

It is well established that a complex inflammatory response to infection plays a significant role in tumorigenesis. In this review, we present a summary of the current resources available in the literature regarding the role of inflammation in anal cancer development.

## 2. Inflammation and Cancer

In 1863, Rudolf Virchow established the first hypothesis about the association between inflammation and cancer based on the presence of leukocytes in neoplastic tissue [6]. Since then, the role of inflammation in tumorigenesis has been extensively investigated. However, its exact impact has not yet been fully elucidated. According to the literature, only 10% of cancers are caused by germline mutations, while others are related to somatic mutations and environmental factors. It is considered that nearly 20% of cancers are associated with chronic infection [7].

Infections, autoimmune diseases, and inflammatory diseases of unknown etiology cause chronic inflammation and an increased risk of tumorigenesis [6]. Chronic inflammation has been associated with different pathophysiological mechanisms that lead to cellular transformation and several changes in immunological response, allowing tumor cells to avoid apoptosis and immune surveillance [8].

These findings highlight the significance of prevention and early detection of infection. Continuous education and implementation of screening programs might result in the prevention of the development of chronic infection and inflammation and thus reduce rates of cancer associated with inflammation.

## 3. HPV Infection and Anal Cancer

Human papillomavirus (HPV) is a double stranded DNA virus that is transmitted through skin or mucosal lesions due to direct skin or mucosal contact with the infected area. HPV infection is considered the most common sexually transmitted infection [5]. According to the available data, the risk of being infected at least once in a lifetime for both men and women is 50% [9]. Risk factors for HPV infection, particularly oncogenic types, are younger age, “men having sex with men” (MSM) population, presence of two or more sexually transmitted infections, and immunosuppression [10].

The established role of HPV infection in cervical cancer pathogenesis has provoked an interest in elucidating its potential role in the pathogenesis of other cancers. Therefore, it has been widely recognized as an important factor in the tumorigenesis of epithelial squamous cell cancers of different locations, especially of the oropharynx and anogenital tract.

HPV infection is recognized as one of the most significant factors included in anal SCC tumorigenesis. According to a recent systematic review, 72% of patients diagnosed with anal cancer were HPV DNA positive [5]. In addition, several studies have reported a presence of HPV infection in 80% to 97% of patients with diagnosed anal cancer [4,11,12,13,14,15].

In women with diagnosed cervical HPV infection, especially caused by high risk HPV types 16 and 18, screening for anal cancer should be considered. However, there is no consensus in guidelines regarding screening for AIN and anal cancer. The decision should be made according to the presence of symptoms, risk factors, concomitant diseases—especially the presence of other HPV-related diseases—and/or medications.

Other established risk factors, such as a high lifetime number of sexual partners and receptive anal intercourse, essentially increase the risk of exposure to HPV, which can lead to persistent infection and potential anal intraepithelial neoplasia (AIN) formation [11]. Drug-induced immunosuppression and immunodeficiency caused by HIV promote anal HPV infection. Exposure to the risk factors has significantly increased over the past few decades due to social, cultural, and lifestyle changes.

At this point, it is important to highlight the influence of HIV stigmatization on seeking medical attention and early detection of other diseases in HIV-infected individuals. According to a recent study [16], HIV-infected women in Nigeria who experienced HIV stigmatization stated that it impeded their cervical screening acceptance. People living with HIV often face stigma and discrimination in their community, but also in healthcare settings. It is important to implement a continuous HIV/AIDS education program for community members and healthcare providers in order to raise awareness of the problems HIV-infected individuals face in everyday life and to enable their appropriate medical treatment.

There are over 200 known HPV strains. Among them, types 6, 11, 16, and 18 are associated with anal infection [4,14]. Besides these types, some researchers discovered a connection between types 31, 33, and 45 and anal cancer [17]. Types 6 and 11 are considered “low risk” strains, as they mainly cause the development of papillomas, whose progression to invasive anal cancer in immunocompetent individuals is uncommon [14,18,19]. However, types 16 and 18 are related to a higher risk of AIN and cancer development [20]. Type 16 is thought to be responsible for up to 75% of invasive anal cancers [4]. The transitional squamocolumnar epithelium is considered the most vulnerable zone to HPV infection [21].

Chronic HPV infection with a high viral load enhances the possibility of AIN development. HPV infects basal epithelial cells and stimulates transcription of a few “early” and “late” viral proteins [22]. E1–E8 are known as “early” viral genes, and they promote viral replication and viral protein translation [14]. Oncoproteins E6 and E7 reduce tumor suppressor protein p53 and the retinoblastoma protein (Rb) activity in epithelial cells, which causes viral genome replication and epithelial cell proliferation. These changes lead to dysplasia, which can progress to invasive anal cancer [14]. Progression of AIN grade III to invasive cancer has been observed in up to 13% of patients over a 5-year period and in about 50% of immunosuppressed patients [23]. Education about sexually transmitted diseases and promotion of vaccination against HPV in both girls and boys are of great importance and should be implemented in everyday clinical practice. It is considered that up to 80% of anal cancers could be prevented by vaccination against oncogenic HPV (against HPV types 6, 11, 16, and 18) [11].

## 4. HIV Infection and Anal Cancer

HIV infection has been established as an independent risk factor for anal cancer. HIV-infected individuals have a 15 to 40–fold higher risk of being diagnosed with anal cancer in comparison to the general population [5,24]. In addition, according to one study, HIV-infected individuals had a 120-fold higher risk when compared to HIV-negative individuals [25]. It is especially common among HIV-infected MSM [26]. According to the current literature, AIN prevalence ranges from 26% to 89% in HIV-infected patients [4]. In addition, a higher rate of AIN progression into invasive cancer has also been reported in HIV-infected individuals, even in those without AIDS [14].

After 1996, the anal cancer incidence increased in comparison to the rates before this period, 75–135 vs. 15 per 100,000 person-years, respectively [5,11], and it is now being recognized as the fourth most common cancer in HIV-infected individuals [27].

Interestingly, highly active antiretroviral therapy (HAART) does not have an effect on the anal cancer incidence rate in this population [28]. Implementation of anal cancer screening programs in HIV-infected individuals has shown no benefit in incidence and mortality reduction [29]. However, some guidelines recommend routine screening for AIN in this population.

## 5. HPV and HIV Coinfection in Anal Cancer

It is well established that HPV infection occurs more often in HIV-infected individuals. According to the current literature, over 90% of HIV positive MSM have HPV coinfection [30]. Furthermore, an increased rate of anal SCC has been observed in patients with HIV and HPV coinfection [31,32].

The immunological and cellular changes due to HPV and HIV coinfection are complex and have not yet been fully elucidated. Previously, it was thought that HIV infection enhances HPV’s oncogenic potential by reducing the immune response. However, recent studies suggest that an increased oncogenic potential is a consequence of microenvironmental changes in the anal epithelium [33]. It is considered that increased risk is not only the result of a low circulating CD4+ cells, because higher rates of anal cancer have been observed among HIV-infected individuals even when their CD4+ cell count is normal.

This observation leads to the conclusion that other pathways are likely involved in the etiology of anal cancer associated with HPV and HIV coinfection. Therefore, recent studies have investigated the role of CD8+ cells and the PD-1/PD-L1 axis [14].

According to the new findings, coinfection occurs locoregionally, in the anal epithelium. HIV impairs epithelial integrity and thus facilitates HPV infection [34]. HIV enters the cells via CD4, CXCR4, and CCR5 receptors [35]. Immune cells with these receptors are especially present in the distal part of the gastrointestinal tract [36]. As a result, anal epithelial cells are exposed to infected immune cells and HIV infection at a local level [14].

The HIV Tat protein is a significant factor included in pathophysiological mechanisms. HIV-infected immune cells secrete the HIV Tat protein, which is believed to enter anal epithelial cells and promote viral DNA transcription and replication [37,38]. HIV Tat protein binds not only RNA polymerase II but also several other proteins involved in the transcription of integrated viral DNA [33,39]. The HIV Tat protein stimulates viral genome replication by enhancing the expression of proteins E6, E7, and E2 [14,40,41]. In addition, it has been discovered that it also lowers p53 levels in cervical carcinoma [42].

Furthermore, HPV infection also facilitates HIV infection. CD4+ cells, dendritic cells, and macrophages are considered HIV target cells. According to studies, the presence of these cells has been reported in HPV-associated anal lesions [43].

## 6. The Role of CD8+ Cells in HPV and HIV Coinfection

CD8+ cells have a role of tremendous significance in the immunologic response to HPV infection. E6- or E7- CD8+ cells elicit apoptosis of HPV-infected cells. The mechanism includes secretion of perforin and granzyme B [44]. However, HIV infection modifies the function of CD8+ cells, causing their reduced activity and efficacy. In one study, HIV coinfection reduced CD8+ cell infiltration of anal precancerous lesions [45]. These changes resulted in impaired viral clearance and progression to HPV-associated precancerous lesions and anal cancer [46]. This finding is supported by other studies, which have shown that increased CD8+ cell tumor infiltration is associated with improved treatment response and a better survival rate [47,48]. In addition, the role of CD8+ cells depends not only on their presence but also on their phenotype and activity [49].

## 7. The Role of PD-1/PD-L1 Axis in HPV and HIV Coinfection

The role of PD-1/PD-L1 axis in anal cancer pathogenesis has been proven. PD-1, or programmed death 1, is an inhibitory receptor located on the T cells’ surface, with a role in controlling their function and proliferation [14]. It interacts with PD- ligand 1 (PD-L1), which can be expressed by regulatory T cells as well as myeloid and tumor cells [14]. In terms of chronic infection and inflammation, increased secretion of cytokines, especially interferon gamma, results in elevated PD-L1 expression, and a reduced immunological response. Tumor cells that express PD-L1 use the same mechanism in order to actively avoid immune destruction [14].

Chronic infection leads to a state of chronic inflammation, which is seen in HIV-infected individuals. Persistent CD8+ cell activation leads to their exhaustion and upregulation of PD-1 expression, which enables PD-L1 expressing tumor cells to avoid their cytotoxicity [50,51,52].

The association between HPV infection and the PD-1/PD-L1 axis has also been reported. Oncoproteins E6 and E7 activate the PD-1/PD-L1 axis [14]. As a result, the local immune response is suppressed [53]. However, there is some inconsistency regarding the association between the PD-1/PD-L1 axis and anal SCC in HPV-infected patients. Some studies have reported that upregulated expression of PD-L1 is associated with the progression of precancerous lesions and, in the case of anal SCC, a reduced survival rate [54,55]. In contrast, it has been observed that PD-L1 expression is associated with improved outcome [14]. This mechanism is notably enhanced in HIV-HPV coinfection, leading to reduced viral clearance, precancerous lesion formation, and cancer development [56]. In addition, one study suggested other mechanisms involved in the pathogenesis of anal cancer in terms of HPV-HIV coinfection. In this study, it was reported that HIV stimulated overexpression of FoxP3 in regulatory T cells, resulting in local dendritic cell exhaustion and a decreased immune response [57].

Further research is needed to elucidate the role of tumor infiltration by CD8+ cells and its association with the PD-1/PD-L1 axis. Those findings could be valuable in implementation and further development of immunotherapy in anal SCC treatment. PD-1 blocking agents could be used in order to improve CD8+ cells’ cytotoxic activity. Currently, there are several clinical trials investigating the role of monoclonal antibodies that block PD-1 receptors in anal SCC [14]. Pembrolizumab and nivolumab have been investigated [5,14,58,59], and the results of these clinical trials could lead to new immunotherapeutic approaches for anal SCC treatment.

## 8. Inflammatory Bowel Disease and Anal Cancer

Crohn’s disease (CD) and ulcerative colitis (UC) are immune-mediated diseases of unknown cause that affect genetically susceptible individuals [2]. These diseases are characterized by lifelong active and inactive phases of disease, requiring a multidisciplinary therapeutic approach that, among others, includes immunosuppressive therapy. Colorectal cancer is more common in CD and UC patients who have severe colitis and/or primary sclerosing cholangitis [2].

## 9. Malignant Transformation of Anal or Perianal Chronic Lesions in CD

In 1975, Lightdale et al. [60,61] first reported on the occurrence of anal cancer in patients with CD anal fistula. According to the meta-analysis [61], the incidence of anal cancer related to CD-fistula was 0.2 per 1000 patient-years, and most likely all of them originated from perianal fistulas. The time between CD onset and cancer diagnosis was 24 and 18 years in men and women, respectively. In female patients with CD-fistula, the time interval for cancer onset was 8.3 years, while this interval was longer in males (16 years) [60,61].

## 10. Adenocarcinoma

The incidence of anal and perianal adenocarcinoma is 0.002 per 1000 patients [60,62,63]. About 50% of cases are primary glandular cancers, mostly arising from perianal lesions of Paget’s disease, and the other half arise from fistulas. In the general population, cancers of the distal rectum and anal canal are only reported in case reports and small case series [60,64,65]. In patients with CD, adenocarcinoma may arise from irreversible chronic anorectal strictures.

## 11. Carcinogenesis and Prognosis of Anal Cancers in Inflammatory Bowel Disease

According to a systematic review, the 5-year survival rate in patients who have anal SCC and inflammatory bowel disease is 37%, while it exceeds 60% in the general population [60,66,67,68]. The pathogenesis of anal cancer in IBD is associated with inflammatory mechanisms that include systemic and local inflammation, HPV infection, drug-induced immunosuppression, and decreased defensive function. Thomas et al. [61] reported that anal cancer was suspected at the first visit and confirmed with biopsy in only 20% of the patients. Clinical examination and biopsy procedures are usually done under anesthesia due to pain and strictures of the anal canal [69]. Even under anesthesia, malignant cells may be missed by superficial biopsies [69,70].

Potential risk for malignant alteration and occurrence of anal canal or perianal cancer in patients with IBD is still unclear. In a systematic review [66], an increased risk was observed in patients with CD, possibly attributable to persistent HPV infection and chronic inflammation [2,71,72]. Since anal CD is mostly associated with inflammation of rectal mucosa and perianal fistula, theoretically, the potential risk of rectal and anal cancer is higher than in the general population. To date, there is no enough evidence for this theory, because IBD phenotype data are lacking in national registries and medical databases.

A recent study by Beaugerie et al. [71] assessed malignancy risk in IBD patients using data collected in the French observational cohort study CESAME (“Cancers Et Surrisque Associe aux Maladies inflammatories intestinalis En France”), where an increased risk for anal and rectal cancer in the IBD subgroup of patients was observed. From May 2004 to June 2005, a total of 19,486 consecutive IBD patients were enrolled in the study. Data relevant to the research were collected electronically. At the beginning of the observation period, 11,759 of the 19,486 enrolled patients had CD, and in 2911 (24.8%), previous or current anal or perianal lesions were found [71]. The remaining 7727 patients had UC or unclassified IBD. In this population, three patients had anal SCC, and eight had rectal adenocarcinoma [71]. The incidence rate for anal SCC was 0.26 per 1000 patient-years and 0.38 per 1000 patient-years for perianal fistula related adenocarcinoma [71]. This study showed a higher incidence of anal and rectal cancer in IBD patients compared to the general population, suggesting that IBD patients may be at an increased risk for anal and rectal cancer development.

## 12. Conclusions

This review summarizes the current available literature regarding the role of inflammation in anal cancer development. The understanding of pathogenesis and risk factors for anal cancer is of great importance, since its incidence rate has increased in the last few decades. The association between inflammation and cancer development is widely accepted, and this review highlights the role of inflammatory processes caused by HPV infection, HIV infection, HPV and HIV coinfection, and inflammatory bowel diseases in anal cancer development. However, there are still many molecular and cellular mechanisms that remain largely unexplored. Further studies on this topic could be of tremendous significance in elucidating anal cancer pathogenesis and could lead to the development of new immunotherapeutic approaches and, eventually, better outcomes.

Even though chemoradiotherapy is considered as a primary therapeutic option, abdominoperineal resection is still performed in some patients. A permanent colostomy reduces the quality of life and represents a significant psychosocial problem. Therefore, prevention and early cancer detection are of great significance. Education, prevention programs, and raising awareness among not only patients but also healthcare providers could lead to a reduction in the incidence rates and early cancer detection. It is also important to identify patients at an increased risk based on established risk factors and to create algorithms and consistent recommendations for their evaluation and follow-up. Therefore, a consensus on this topic is necessary.

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
