# Peer review of "The Role of Inflammation in Anal Cancer"

_diseases, 2022, doi:10.3390/diseases10020027_

Round 1

Reviewer 1 Report

This is a comprehensive review of anal cancer and the inflammation. It clearly discusses the role of inflammation in the emergence of anal cancers. The roles of HPV infection, HIV infection, co-infection of HPV and HIV, inflammatory bowel disease are addressed. 

Reviewer 2 Report

Many thanks for the opportunity to review this paper. Below are my comments for the authors:

  1. In the abstract, the authors should consider introducing the aim of their study at the beginning of the abstract rather than at the conclusion.
  2. Based on number 1 comment, the authors should make their conclusion to reflect their suggestions for future studies. 
  3. Point 3 (HPV infection and anal cancer), the first and second paragraphs will also benefit from citing a recent study, see reference: Ogueji, I. A., & Adejumo, A. O. (2022). Perceived HIV stigmatization and association with cervical screening adoption among HIV-positive women in a Nigerian Secondary Health Facility: Implications for psychological interventions. Journal of HIV/AIDS & Social Services, 1-10. https://doi.org/10.1080/15381501.2021.2006104
  4. The entire paper reads like an accumulation of the literature without communicating what are the implications of findings, what are the recommendations for practice, and what are the directions for future research. The authors should consider revising their manuscript to reflect these gaps and should clearly indicate where such revisions are made. I do believe that if the author revise their manuscript accordingly, it could be useful for directing the paths of future research. It will also enhance the scientific soundness and significance of their manuscript. 
  5. The authors should sincerely proofread their entire manuscript for spelling/grammatical errors. 

Best wishes!

Reviewer 3 Report

The authors have sufficiently elaborated and critically reviewed the most relevant literature in their chosen topics, i.e., the critical relationship between inflammation (especially HPV infection elicited) and the anal cancer. 

The topic of this review is of high interest to the field of cancer biology and therapy, especially in anal cancer.

Round 2

Reviewer 2 Report

Thank you to the authors for making the requested changes to their manuscript. I have attached the full paper to their number 16 reference as the authors requested for a full access to the paper to enhance the argument of their manuscript. I suggest that the number 16 reference is also suitable for citation in the first and second paragraphs of the section "HPV infection and anal cancer" (as mentioned in the previous round of peer-review). I recommend that the manuscript be accepted with minor corrections (that is, after the authors have utilized the attached full access paper they requested for). 

Best wishes to the authors!
